# Physical Function, Cognitive Function, and Daily Activities in Patients Hospitalized Due to COVID-19: A Descriptive Cross-Sectional Study in Sweden

**DOI:** 10.3390/ijerph182111600

**Published:** 2021-11-04

**Authors:** Alexandra C. Larsson, Annie Palstam, Hanna C. Persson

**Affiliations:** 1Department of Clinical Neuroscience, Institute of Neuroscience and Physiology, Sahlgrenska Academy, University of Gothenburg, 41345 Gothenburg, Sweden; annie.palstam@gu.se (A.P.); hanna.persson@neuro.gu.se (H.C.P.); 2Department of Occupational Therapy and Physical Therapy, Sahlgrenska University Hospital, 41346 Gothenburg, Sweden; 3Department of Neuro Science, Sahlgrenska University Hospital, 41346 Gothenburg, Sweden

**Keywords:** COVID-19, rehabilitation, recovery of function, functional status, activities of daily living, cognition, hospitals

## Abstract

An estimated 14–20% of people infected with COVID-19 require medical care. The aim of the present study was to evaluate physical function, cognitive function, and daily activities in patients hospitalized due to COVID-19, and to investigate differences depending on age and admission to the intensive care unit (ICU). This prospective descriptive cross-sectional study included a consecutive sample of 211 patients (mean age 65.1 years, 67.3% men) hospitalized due to COVID-19 in Sweden. Data regarding physical function and daily activities were collected in hospital from July 2020 to February 2021. The average length of hospital stay was 33.8 days, and 48.8% of the patients were admitted to the ICU. Physical function (grip- and lower body strength) was reduced in both groups, and significantly more in the older group, ≥65 years old, compared to the younger. Furthermore, the older group also had significantly less ability to perform activities in daily life, and had significantly reduced cognitive function as compared to the younger age group. In patients treated in the ICU, physical impairments as well as the activity level were significantly more pronounced compared to patients not treated in the ICU. Patients hospitalized due to COVID-19 are physically impaired, have mild cognitive impairments, and have difficulties performing daily activities. The findings in this study indicate the need for out-patient follow-up and rehabilitation for patients hospitalized due to COVID-19, especially in older patients and patients treated in the ICU.

## 1. Introduction

SARS-CoV-2 causes COVID-19, affects society as a whole, can involve suffering for both affected individuals and their families, and is a great challenge for healthcare and economic systems all over the world [1]. Considering the load on national healthcare systems during the pandemic outbreak, defining the precise settings and methods of intervention is important to optimize the rehabilitation process and lessen the suffering for the individual [1]. Sweden chose a different strategy than neighboring Nordic and European countries [2], and long-term outcomes of different strategies are not yet fully uncovered.

Symptoms of COVID-19 vary, as does severity and extent of impairments over time [1,3,4]. Acute symptoms of COVID-19 can include fever, cough, myalgia, and fatigue [5]. An estimated 14–20% of COVID-19 patients require medical care [5,6]. More than 75% of patients hospitalized with COVID-19 require supplemental oxygen [4], and 3–5% develop acute respiratory distress syndrome (ARDS), including the need for mechanical ventilators, and need to be admitted to an intensive care unit (ICU) [4,5,6]. Prolonged hospital stay, duration of mechanical ventilator use, and ICU care may lead to critical illness, such as post-intensive care syndrome (PICS) and ICU-acquired weakness (ICU-AW) [6,7,8]. Cognitive impairments also seem to be a consequence of COVID-19 [9,10,11]; the cognitive malfunctioning after hospitalization due to COVID-19 has been shown to be associated with length of stay in the ICU [12].

Gradually, the pressure on ICUs has decreased and there has been a shift of focus towards the needs for COVID-19 rehabilitation [2]. The full needs for rehabilitation after COVID-19 are not yet known, nor is how physical and cognitive function during hospitalization are affected and recovered in individuals with severe COVID-19. Rehabilitation needs have been shown during all phases of COVID-19; therefore, rehabilitation professionals should be positioned in the ICUs, hospital wards, and step-down facilities [6]. The rehabilitation process after intensive care can be long and extensive [6,7,8], and rehabilitation needs after severe COVID-19 may be amplified by underlying health conditions, comorbidities, and older age [6,13].

Furthermore, in previous patients infected with SARS-CoV (i.e., SARS), physical function and fitness were reduced after ARDS and patients had incomplete recovery of physical function, as well as long-term impairments 1 to 2 years post-infection [14]. Considering the similar pathology of COVID-19 and SARS, patients can be anticipated to manifest comparable impairments in functioning [14]. Though survival is high in critically ill COVID-19 patients, the prolonged need for mechanical ventilation may result in intensive care-acquired weakness [15,16]. The long-term effects of COVID-19 are currently unknown, but it is probable that patients with severe illness are likely to suffer substantial sequalae [4]. Although patients’ strength has been shown to improve during hospitalization, the impact on functioning has been shown to remain substantial [15]. It has been shown that the majority of patients treated in the ICU are functionally dependent at discharge [17]. Elderly patients seem to be particularly susceptible to more adverse clinical outcomes of COVID-19 [13]. A few studies have investigated the level of physical functioning at hospital discharge [17,18,19,20,21,22], mainly presenting data from the first wave and none in a Swedish cohort. The international classification of disability and health, ICF, describes different aspects of health conditions such as body functions (physical as well as cognitive functions) and activities [23]. The ICF has been described as useful to capture the complexity of different symptoms that patients with COVID-19 may have [24].

The aim of the present study was to evaluate physical function, cognitive function, and daily activities in patients hospitalized due to COVID-19 during the first and second wave and investigate differences based on age and ICU care in a well-defined geographic catchment area in Sweden.

## 2. Materials and Methods

### 2.1. Patients

This is a regional, hospital-based, cross-sectional study presenting data from the first assessment in the longitudinal study Life in the time Of COVID study in Gothenburg (GOT-LOCO). With an intention of consecutive inclusion, patients were included from five hospitals in the Västra Götaland region (VGR) with a catchment area of 1.67 million, corresponding to 16% of the population in Sweden. Data were collected from 9th of July 2020 to 23rd of Febuary 2021 at Sahlgrenska University Hospital (SU) (four units: Östra Hospital, Mölndal Hospital, Högsbo Hospital, and Sahlgrenska Hospital), Södra Älvsborgs Hospital, Skaraborgs Hospital, Alingsås Hospital, and NU healthcare (two units: Norra Älvsborgs Hospital and Uddevalla Hospital). The study was approved by the Swedish Ethical Review Authority (Dnr: 2020-03046, 2020-03922), complies with the declaration of Helsinki, and is structured in accordance with STROBE guidelines. The inclusion criteria were: patients with COVID-19 who were admitted to hospitals within the VGR and were non-contagious when enrolled, had an expected hospital care period ≥5 days, were ≥18 years old, and previously lived in own housing. Patients were excluded if they were unable to provide informed consent or if comorbidity indicated high 1-year mortality (i.e., palliative care or metastatic cancer), and if they were not Swedish residents.

Eligible patients were identified by the study coordinator or by a local test leader (a physical or occupational therapist working at the hospital) at each hospital. If needed, the inclusion process was discussed with a physician within the research group, in order to avoid selection bias. All test leaders and physical and occupational therapists involved in the study at each hospital were trained by the study coordinator to ensure a standardized inclusion process and data collection. Data collection followed a pre-specified procedure and was conducted by a physical therapist and an occupational therapist before the patient was discharged. Eligible patients were informed, and consent was obtained prior to data collection.

### 2.2. Data Collection

Data collection included variables retrieved from medical charts concerning length of hospital stay and ICU care; level of oxygen saturation at arrival and during the hospital stay; comorbidities; height, weight, and body mass index (BMI); and whether the patient was re-admitted to the hospital within the study period. The length of hospital stay was calculated from the day the patient was admitted to the day of discharge, which was also valid if the patient was infected with COVD-19 during the hospital stay. For the patients who were “in and out from the hospital” (e.g., discharged home or to a short-term nursing home, pending on in-hospital rehabilitation), the total amount of days at the hospital was calculated. To describe the study sample and the patients’ characteristics at hospital discharge, the Charlson Comorbidity Index (CCI) was collected from the patients’ medical charts. The weighted index according to Sundararajan et al. [25] was used.

Aspects of physical function were evaluated using performance-based tests. The 30-s chair rise test was used to evaluate lower body strength [26]. The patient was instructed to rise from a chair and sit back down as many times as possible during 30 s with their arms crossed over their chest [26]. If the patient needed to use the arm rest or needed other assistance, this was noted, and those patients were registered as “0” chair rises. Walking capacity was evaluated with both comfortable gait speed (CGS) and fast gait speed (FGS) and assessed with the 10-m walking test (10MWT) [27] on a flat surface, and data comprised of the number of steps and seconds. Heart rate and oxygen saturation (SpO2) were noted before and after each test, as well as whether the patient used any walking aid. Grip strength [28] (kilograms) was measured using a JAMAR Hand Dynamometer (Sammsons Preston, Chicago), a sealed hydraulic hand dynamometer. The procedure was repeated in three trials in each hand, respectively, and the mean of the three trials was calculated as well as the percentage of normative values.

Cognitive function was assessed with the Montreal Cognitive Assessment (MoCA), which is a brief screening test for mild cognitive impairment, with a maximum score of 30 points, where scores below 26 points indicate cognitive impairment [29]. The MoCA has high reliability and validity in patients with mild cognitive impairment [29] and has been previously used to evaluate cognitive function in patients hospitalized due to COVID-19 [30] and patients recovering from COVID-19 [9]. To evaluate visual search, scanning, speed of processing, mental flexibility, and executive functions [31], trail making tests A and B (TMT A, TMT B) were used. TMT B is the more demanding test, involving further demands on the patient’s cognitive flexibility [31].

The patient’s ability to perform daily activities was evaluated using the Barthel Index (BI) [32], which has a range of 0–100, where scores below 95 points indicate impaired ability to perform daily activities [33]. Functional ambulation category (FAC) was used to evaluate the patient’s ability to ambulate (range: 0–5) [34]. The maximum score of 5 indicates that a person can ambulate independently in society. Post-COVID functional status (PCFS) is a self-assessed questionnaire developed to detect functional status post-acute COVID-19 infection and to evaluate the consequences of the infection [35], and this summarizes the patient’s views on impairment after COVID-19 infection. The patients estimate their physical impairments due to COVID-19 with the main subject being if they can manage to live at home without any assistance after the infection. Scores range from 0–4, with 4 being totally physically impaired and 0 not feeling affected at all.

### 2.3. Data Analysis

The data were processed and analyzed using SPSS Statistics 27 (IBM Corporation, Armonk, NY, USA). Descriptive statistics using the number and percentage, mean and standard deviation (SD), or median and interquartile range (IQR) were carried out for the total population, as well as for groups based on age and ICU admission. Younger patients correspond to those <65 years old and the older patients to those ≥65 years old. Differences between groups were calculated with the independent sample *t*-test for continuous variables and the Mann–Whitney U test for ordinal variables. The level of significance was set to *p* < 0.05.

## 3. Results

The study population comprised 211 patients, including 142 men (67.3%). The mean age was 65.1 years (SD 13.4) and the average BMI was 29.11 (SD 7.4 kg/m^2^). The majority (54%, n = 114) of the population was ≥65 years old (Table 1). Data collection was performed by physical therapists in the median of 1 day (IQR 7) and by the occupational therapist in median 2 days (IQR 8) prior hospital discharge. The patients were discharged from the hospital after a mean of 33.8 days (SD 35.6), and no difference in length of stay was found depending on age (*p* = 0.691). Of the study population 48.8% were admitted to the ICU, with a mean ICU length of stay of 18.5 days (SD 19). The length of ICU stay did not differ between the older and younger age groups (*p* = 0.749, Table 1).

The older group had significantly lower physical function than the younger group in regards to lower body strength (*p* = 0.001) and grip strength (right hand, *p* = 0.001; left hand, *p* = 0.005). These physical impairments were also seen in patients treated in the ICU compared to the patients not treated in the ICU (lower body strength: *p* = 0.003; grip strength: right hand, *p* = 0.001 and left hand, *p* = 0.003).

Of the 167 patients who performed the 10MWT test, only 56.3% managed to do so without any walking aid. A walker was the most commonly used walking aid, used by 32.3% of the population. Directly after the test, patients had decreased in SpO2 with an average of 1.4%. Desaturation was also seen after performing the 30 s chair rise test, with patients averagely decreasing—3.0% SpO2.

Cognitive function assessed with the MoCA was below the cutoff of cognitive impairment with a median score of 25 points (IQR 6) for the total study population (Figure 1). The older patients demonstrated lower cognitive function than the younger patients (MoCA, *p* = 0.001; TMT A, *p* = 0.001; TMT B, *p* = 0.001). There was no significant difference in cognitive function between the patients depending on ICU admission (Table 2 and Figure 1).

The ability to perform daily activities was significantly lower in the older age group compared to the younger age group (BI, *p* = 0.001; FAC, *p* = 0.00; PCFS, *p* = 0.012). Patients treated in the ICU also presented significantly more activity limitations than patients not treated in the ICU (BI, *p* = 0.001; FAC, *p* = 0.001; PCFS, *p* = 0.001; Table 2).

## 4. Discussion

This is a prospective study of 211 patients hospitalized due to COVID-19, with data collected from the first, as well as the second, wave in a well-defined geographic catchment area in Sweden. Patients hospitalized with COVID-19 presented with physical impairments, mild cognitive impairments, and activity limitations. The impairments were more pronounced in patients ≥65 years old and in patients who had been admitted to the ICU. However, the heterogeneity within the cohort was large in the domains measured.

Lower body strength and grip strength were greatly reduced in the present sample, where the lower body strength of the population was comparable to the performance of 94+-year-olds in a healthy population [36], although the mean age of the study sample was 65 years, indicating high risk for mobility problems and risk for falls [36].

In addition, only 114 patients managed to perform the 30-s char rise test without assistance from the arms, indicating that the weakness was even more pronounced (Table 2). This also verifies findings that the impairment of COVID-19 are not only respiratory, but can also lead to impairments of motor functions [19]. The reduced strength at the time of assessment could also be due to long hospitalization in isolation, which may have resulted in general fatigue, prolonged bed rest, inactivity, and lack of motivation when performing the test. Furthermore, chair stand performance is not only associated with strength, but also with balance and psychological status [37]. The ability to perfom chair stands is likewise asociated with the ability to care for one self and managing to live independently [36], highlighting the importance of further rehabilitation for these patients. The patients treated in the ICU had a significantly lower level of physical function than patients treated in general hospital wards. This study supports the findings of previous studies regarding physical function in ICU-treated patients [7], confirming that this also applies to patients with COVID-19. Keepig in mind that the mean age of the ICU patients was 62.9 years, a substantial part of the population will return to work and everyday life with reduced strength.

In the present study, the population presented mild cognitive impairment when screening with the MoCA, with a median score of 25. This is in line with previous studies showing that COVID-19 could affect cognitive function [9,10,11]. One small study found that 61.5% demonstrated cognitive impairment in the acute phase of COVID-19 [11], whereas another study on hospitalized COVID-19 patients suggested that the impairments persisted after hospital discharge [38]. Why some patients with COVID-19 manifest cognitive impairment is not known, but it could be due to the development of ARDS or infection in the central nervous system (CNS) [11]. The results from the current study indicate a prevalence of cognitive impairment in the cohort; however, the results should be interpreted with caution. Cognitive screening assessment in the early phase or during the hospital stay could have been influenced by several factors, such as fatigue, stress, or lack of motivation. The older patients were more cognitively impaired than the younger patients (Table 2). This finding is in line with what might be expected in an elderly population and what was seen in a previous study of COVID-19 patients [30], as well as what has been described by the Pan American health organization [6]. Although the total study sample scored below the 26-point cutoff on the MoCA, no difference in the level of cognition was seen based on admission to the ICU. Previous studies have described cognitive impairments in ICU-treated patients due to COVID-19 [17,30,38]. The differences between studies may be due to different time points of assessment during the hospital stay. Older patients may manifest cognitive impairment at hospital discharge after COVID-19 infection and should be assessed for cognitive impairment when referred for follow-up in primary care.

The impaired physical and cognitive functions also impacted the patients’ abilities to perform daily activities. The PCFS scores indicated that the patients themselves expressed difficulties managing daily activities, although they were able to take care of themselves. For both the ICU treated patients and the older group, the BI was in line with what has been previously shown in hospitalized COVID-19 patients, being somewhat dependent after hospitalization [17,18,22,30]. In particular, this suggests that older patients and ICU-treated patients require further follow-up after hospital discharge.

Implementing a clinical study in the middle of a pandemic presented some difficulties. This study presented physical and occupational therapists in the clinics with a large test battery of clinical outcome measurements. This, together with local deviations in the clinical experience, somewhat limited the selection of outcome measurements. For example, the present study could not implement a specific evaluation of respiratory function, which is a common complication of COVID-19 [4]. Since respiratory function often is affected by COVID-19 [4], future studies should not only focus on SpO2 at hospital discharge, but may also evaluate respiratory functions by spirometry in order to enable proper follow-up for patients that might suffer from impaired lung function**.** The study sample size was chosen accordingly to address the heterogenicity in symptoms in the study population; however, this was an estimate. Due to the pandemic, no pilot study or knowledge regarding sample size was available. The internal missing data were mostly due to heavy workload at the clinics, or due to restrictions in patient ambulation. The criteria for patients to be considered as non-contagious varied over time with updated guidelines, and between different hospitals. In this study, we included patients that were diagnosed with COVID-19 in their medical charts. How the patients were diagnosed varied over time and differed between the clinics.

Since data were collected from a large catchment area, these results could be anticipated to be generalizable for countries with similar helatchare systems; however, given the rapidly developing knowledge regarding symptoms and care due to COVID-19, this is difficult to evaluate. The study was specifically designed to evaluate physical function, cognitive function, and daily activities in patients hospitalized due to COVID-19. The chosen measurements were carefully selected to specifically evaluate these aspects.

## 5. Conclusions

Patients hospitalized due to COVID-19 are physically impaired, have mild cognitive impairments, and have difficulties performing daily activities at discharge. Impairments are more pronounced in patients treated in the ICU and patients over 65 years of age. The findings indicate the need for out-patient follow-up and rehabilitation for patients hospitalized due to COVID-19, especially for older patients and patients treated in the ICU. More research is needed to follow up on how physical and cognitive impairments and the ability to perform activities develop over time.

## Figures and Tables

**Figure 1 ijerph-18-11600-f001:**
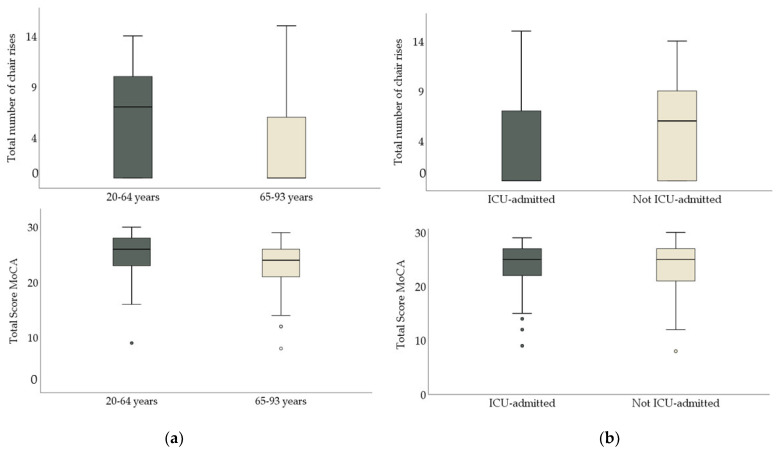
(**a**) To the left: total number of chair rises and total scare of Montreal Cognitive Assessment Scale (MoCA), depending on age group. (**b**) To the right: total number of chair rises and total score of MoCa depending on ICU–care.

**Table 1 ijerph-18-11600-t001:** Patient characteristics.

	All Patients (n = 211)	Under 65 Years (n = 97)	Over 65 Years (n = 114)	ICU Admitted (n = 104)	Non-ICU (n = 107)
Age, years	65.1 ± 13.4	53.4 ± 8.90	75.1 ± 6.86	62.9 ± 11.85	67.1 ± 14.47
Male	143 (67.8)	64 (66)	79 (69.3)	80 (76.9)	63 (58.9)
BMI, kg/m^2^ n = 138	29.1 ± 7.4	30.3 ± 7.7	28.1 ± 7.0	30.0 ± 7.2	28.1 ± 7.6
Total LOS, days	34.1 ± 35.8	33.9 ± 42.3	35.0 ± 29.9	52.2 ± 42.3	17.3 ± 15.0
ICU admission	104 (49.3)	53 (54.6)	51 (44.7)		
- Intubated	64 (66)	33 (35.1)	31 (28.2)		
- LOS, days	18.6 ± 18.9	19.11 ± 21.8	18.04 ± 15.5		
SpO2% at admission, n = 199	85.8% ± 11.7	84.9% ± 12.5	86.6% ± 10.8	83.4% ± 14.2	88.3% ± 7.7
NIV / HFNC *n* = 204	128 (62.7)	64 (68.8)	64 (57.7)	91 (92.9)	36 (34.3)
Pulmonary embolism n = 207	24 (11.6)	10 (10.6)	14 (12.4)	14 (13.5)	10 (9.7)
Venosus thromboembolism n = 206	4 (1.9)	1 (1.1)	3 (2.7)	2 (1.9)	2 (1.9)
Pneumonia, n = 188	44 (23.4)	19 (22.4)	25 (24.3)	33 (35.1)	11 (11.7)
Charlson Comorbidity					
Index at discharge, median (min-max) n = 209	1 (0–7)				
0 points	66 (31.6)	42 (43.8)	24 (21.2)	30 (29.1)	36 (34)
1–2 points	95 (45.5)	39 (40.7)	56 (49.6)	46 (44.6)	49 (46.2)
3–4 points	36 (17.2)	11 (11.4)	25 (22.1)	20 (19.5)	16 (15.1)
≥5 points	12 (5.7)	4 (4.1)	8 (7.1)	7 (6.8)	5 (4.7)
Discharged to n = 209					
- Home	153 (73.2)	87 (90.6)	66 (58.4)	81 (78.6)	72 (67.9)
- Home with nursing assistance	33 (15.8)	8 (8.3)	25 (22.1)	15 (14.6)	18 (17)
- Short-term nursing home	20 (9.6)		20 (9.6)	6 (5.8)	14 (13.2)
- In and out before finally discharged	2 (1)	1 (1)	1 (0.9)	1 (1)	1 (0.9)
- Diseased before discharge	1 (0.5)		1 (0.9)		1 (0.9)
- Stay in specialized rehabilitation setting before discharge	51 (24.2)	26 (26.8)	25 (21.9)	41 (39.8)	10 (9.3)
- Re-admitted patientsn =186	20 (10.8)	8 (9.6)	12 (11.7)	8 (8.6)	12 (12.9)

Data are given as mean ± standard deviation or n (%) unless otherwise noted. Abbreviations: BMI: body mass index; SG-PALS: Saltin Grimby-physical activity level scale; ICU: intensive care unit; LOS: length of stay; SpO2: peripheral oxygen saturation; NIV: non-invasive mechanical ventilation; HNFC: high flow nasal canula.

**Table 2 ijerph-18-11600-t002:** Results and group comparisons.

	All Patients (n = 211)	Under 65 years (n = 97)	Over 65 years (n = 114)	*p*-Value *	ICU Admitted (n = 104)	Non-ICU (n = 107)	*p*-Value *
Body function							
Lower body strength							
30-s chair stand test n =194 ^a^	4.7 ± 4.6	6.5 ± 4.7	3 ± 3.8	** *0.001* **	3.7 ± 4.5	5.7 ± 4.5	** *0.003* **
Grip strength							
JAMAR Right hand, kilos n = 192	27.7 ± 14.6	31.8 ± 15.9	24.2 ± 12.5	** *0.001* **	24.2 ± 13	31 ± 15.3	** *0.001* **
JAMAR, Left hand, kilos *n = 192*	26 ± 13.6	29.1 ± 14.5	23.5 ± 12.4	** *0.005* **	23.1 ± 12.7	29 ± 14	** *0.003* **
Walking capacity							
10 MWT, CGS, seconds n = 167	16.1 ± 7.9	14.6 ± 8.3	17.4 ± 7.3	** *0.022* **	17.2 ± 8.4	15 ± 7.2	*0.066*
10 MWT, CGS, steps n = 166	21.6 ± 6.6	20.1 ± 6.2	22.9 ± 6.7	** *0.006* **	22.1 ± 6.2	21.1 ± 7	*0.332*
10 MWT, FGS, seconds n =160	10.9 ± 5.4	10.4 ± 6.5	11.4 ± 4.1	*0.256*	11.6 ± 6.6	10.2 ± 3.9	*0.097*
10 MWT, FGS, steps n =160	17.8 ± 5.8	17.2 ± 6.2	18.4 ± 5.3	*0.173*	18.1 ± 6.3	17.5 ± 5.1	*0.557*
Cognitive function							
MoCA n = 176	25 (6)	26 (5)	24 (6)	** *0.001* **	25 (6)	25 (6)	*0.551*
TMT A n = 182	42 (32.8)	35 (30)	51 (43)	** *0.001* **	42.9 (33)	40 (34)	*0.637*
TMT B n =134	105 (87)	82 (66.7)	127 (97.5)	** *0.001* **	100 (89)	105 (70)	*0.846*
Activity							
Barthel Index n = 201	90 (35)	97.5 (20)	85 (40)	** *0.001* **	80 (35)	100 (15)	** *0.001* **
FAC n = 206	4 (2)	5 (1)	4 (1)	** *0.001* **	4 (1)	4.5 (7)	** *0.001* **
PCFS n = 207	3 (2)	3 (2)	3 (2)	** *0.012* **	3 (2)	3 (2)	** *0.001* **

Data are given as mean ± standard deviation or median (interquartile range). * *t*-test or Mann–Whitney U test. ^a^ *n* = 79 patients did not perform the test with arms crossed, needing armrests or other assistance. Abbreviations: CGS: comfortable gait speed; FGS: fast gait speed; MoCA: Montreal Cognitive Assessment Scale; FAC: functional ambulatory category; PCFS: post-COVID functional status. Significant values are marked in bold.

## Data Availability

The datasets analyzed during the current study are not publicly available due to ethical restrictions. According to the Swedish regulation https://etikprovningsmyndigheten.se/ (accessed on 5 October 2021) the permission to use data is only for what has been applied for and then approved by the Swedish Ethical Review Authority. Data are available from the authors (contact Hanna C Persson, hanna.persson@neuro.gu.se) upon reasonable request.

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
