# Peer review of "Physical Function, Cognitive Function, and Daily Activities in Patients Hospitalized Due to COVID-19: A Descriptive Cross-Sectional Study in Sweden"

_ijerph, 2021, doi:10.3390/ijerph182111600_

Round 1

Reviewer 1 Report

STRUCTURE

  • The manuscript is properly structured

TITLE AND ABSTRACT

  • It is preferable not to use the same words in the title and in the abstract.

INTRODUCTION

  • Explain the scientific background in more depth
  • The study presents an approach to obtain original information; however, the introduction is very scarce. More information is needed on the disease, the population it affects, which population is of interest for this study, the sequelae mentioned, what is meant by cognitive and physical function, etc.
  • State specific objectives
  • There are no hypotheses

MATERIAL AND METHODS

Study design

  • Present key elements of study design early in the paper

Setting

  • Describe periods of recruitment and follow-up

Variables

  • Define clearly all exposures, predictors, potential confounders, and effect modifiers

Bias

  • Describe any efforts to address potential sources of bias

Study size

  • How sample size was determined?

Quantitative variables

  • Explain how quantitative variables were handled in the analyses. Describe which groupings were chosen and why

Statistical methods

  • Describe all statistical methods, including those used to control for confounding
  • Explain how missing data were addressed
  • Describe any sensitivity analyses

RESULTS

  • Consider to use a flow diagram. Report numbers of individuals at each stage of study—eg numbers potentially eligible, examined for eligibility, confirmed eligible, included in the study, completing follow-up, and analysed
  • Give reasons for non-participation at each test
  • Give characteristics of study participants (eg demographic, clinical, social) and information on exposures and potential confounders
  • Consider explain the meaning of the bold letter in table 2

DISCUSSION

  • Add reference at line 247
  • The present study's findings can be further discussed with the other research mentioned
  • Discuss the generalisability (external validity) of the study results

REFERENCES

References follow the indicated style

Author Response

Thank you for valid comments regarding this manuscript. We have adressed the comments in point-by-point response in the attached file.

Reviewer 2 Report

Overall, a sound research paper.

Area for improvement: in the introduction, there is not enough context-specific information, for example, comments looking at Sweden more specifically could help the user understand the geographical relevance. Also, mentioning Sweden's policies, given they are markedly different to other parts of the world, aids with interpretation.

Another area of weakness is the lack of visual data presentation. The tables are important but are lists of numbers that require visual translation. Please consider carefully selecting one or two figures to represent the key data.

Author Response

Thank you comments regarding this manuscript. We have provided a point-by-pont response in the attached file.

Reviewer 3 Report

This prospective descriptive cross-sectional study analyses an interesting aspect of the current pandemic, the physical and cognitive impairment following the hospitalization due to COVID-19.

The aims and objectives of the study are clear. The study design seems appropriate for the aim, however, the sample size was not discussed; the target population was clearly defined, then the sample population was defined as a consecutive sample of 211 subjects, but only in the abstract and seems to be taken from an appropriate population in order to represent the source population correctly. The selection process does not seem to have been duly described and does not allow the reviewer to assess if the subjects were representative of the target population, plus the non-responders, one of the main issues in cross-sectional studies, are not even mentioned. The case definition should be more clearly defined, including how the diagnosis of COVID-19 and the negativization (swab and then PCR?) of the infection were assessed.

The risk factor and outcome variables were in most cases measured through exams or instruments that are universally recognized as reliable for the scope of the study; however, the authors underline that the outcome measurements were limited by the pandemic and that more specific evaluations of respiratory function did not take place. This should be emphasized and the value of such measurement in other studies underlined. The statistical methods used are described and the way of expressing statistical significance was specified as the p-value, but no confidence intervals were reported.

The basic data were described in the results, the discussion and conclusion are justified by the results and the weaknesses of the study were mentioned.

There were no problems concerning funding or written consent.

Author Response

Thank you for the comments regarding this manuscript. A point-by-point response is found in the attached file.

Round 2

Reviewer 1 Report

Thank you for your response. Please, review now the references, because do not follow the IJERPH guidelines.